# Identification of the Degree of Degradation of Fibre-Cement Boards Exposed to Fire by Means of the Acoustic Emission Method and Artificial Neural Networks

**DOI:** 10.3390/ma12040656

**Published:** 2019-02-21

**Authors:** Krzysztof Schabowicz, Tomasz Gorzelańczyk, Mateusz Szymków

**Affiliations:** Faculty of Civil Engineering, Wrocław University of Science and Technology, Wybrzeże Wyspiańskiego 27, 50-370 Wrocław, Poland; krzysztof.schabowicz@pwr.edu.pl (K.S.); mat.szymkow@gmail.com (M.S.)

**Keywords:** fibre-cement boards, non-destructive testing, acoustic emission, degree of degradation

## Abstract

This paper presents the results of research aimed at identifying the degree of degradation of fibre-cement boards exposed to fire. The fibre-cement board samples were initially exposed to fire at various durations in the range of 1–15 min. The samples were then subjected to three-point bending and were investigated using the acoustic emission method. Artificial neural networks (ANNs) were employed to analyse the results yielded by the acoustic emission method. Fire was found to have a degrading effect on the fibres contained in the boards. As the length of exposure to fire increased, the fibres underwent gradual degradation, which was reflected in a decrease in the number of acoustic emission (AE) events recognised by the artificial neural networks as accompanying the breaking of the fibres during the three-point bending of the sample. It was shown that it is not sufficient to determine the degree of degradation of fibre-cement boards solely on the basis of bending strength (*MOR*).

## 1. Introduction

The fibre-cement board is a building material that has been used in construction since the beginning of the last century. The Czech engineer Ludwig Hatschek developed and patented the technology for the production of this composite material. The first boards contained asbestos fibres. After asbestos had been found to be carcinogenic, they were replaced with cellulose fibres and synthetic fibres [1]. Being composed of cement, cellulose fibres, synthetic fibres and various innovative additives and admixtures, the currently produced fibre-cement boards are a completely different building product. The added components and fillers of fibre-cement boards are: limestone powder, mica, perlite, kaolin, microspheres and recycled materials [2,3]. As a result, fibre-cement boards continue to be innovative products consistent with the principles of sustainable development [4]. Fibre-cement boards are used in construction mainly for ventilated façade cladding [5], as shown in Figure 1. In the course of their service life, fibre-cement boards are exposed to variable environmental factors, chemical aggressivity (acid rains), and physical aggressivity (ultraviolet radiation). Moreover, fibre-cement boards are exposed to accidental operational factors, mainly the high temperature that a building fire generates.

After a building fire occurs, it is necessary to determine the impact of the high temperatures on the façade cladding of the building and the neighbouring buildings with regard to their further service life. This means that identifying the degree of degradation of fibre-cement boards caused by the impact of fire is vital from both the scientific and practical point of view. Most of the research on fibre-cement boards to date has been limited to determining their standard physicomechanical properties, the effect of operational factors, such as soak–dry cycles, freeze–thaw cycles, heating, raining and high temperatures, and the effect of the use of various types of fibres and production processes, solely through bending (*MOR*) tests [6]. In the literature on the subject, one can find only a few nondestructive tests carried out on fibre-cement boards, limited to the imperfections arising during production [7,8,9,10]. The effect of fire is one of the most destructive accidental factors for many building products, especially the composite ones containing reinforcement in the form of fibres of various kinds, particularly cellulose fibres, considering that at a temperature above 200 °C cellulose fibres undergo pyrolysis. Thus, the action of fire critically affects the durability of the whole composite [6]. In order to verify this hypothesis, experiments were conducted during which fibre-cement board samples were exposed to fire at a temperature of about 400 °C at various durations in the range of 1–15 min. This time range and the temperature of 400 °C had been experimentally determined through many preliminary trials. The aim of the experimental research is to observe the advance of the degradation of the fibres due to fire and the associated high temperature. Preliminary tests conducted at the temperature of 500 °C very quickly, i.e., after 1–2 min, resulted in the complete destruction of the board, which no longer could be subjected to the bending test. A reduction in the temperature to 250–300 °C would considerably lengthen the duration of the tests. Preliminary investigations showed that, at a temperature of 250 °C, the fibres in the boards would be completely destroyed after 2–3 h. Therefore, in the authors’ opinion, the choice of the temperature of 400 °C was optimal for conducting experiments aimed at identifying the degree of degradation of the fibres contained in fibre-cement boards under the high temperature generated by fire.

After they had been exposed to fire, the board samples were subjected to three-point bending, during which the acoustic emission (AE) was recorded, and the results were then analysed by means of artificial neural networks. Research carried out by the authors has shown that the identification of the degree of degradation of fibre-cement boards exposed to fire, based on only bending strength (*MOR*), is inadequate [11,12]. Thanks to the use of the acoustic emission method, the degree of degradation can be determined on the basis of not only the mechanical parameters, but also the acoustic phenomena that can occur in fibre-cement boards. From the recorded AE signals, model (reference) characteristics of the acoustic spectra that accompany the breaking of the cement matrix and the fibres during bending were derived. Under the continuing exposure to fire, the fibres in the boards would undergo gradual degradation, which was reflected in a decrease in the counts of AE events identified as accompanying fibre breaking. This phenomenon is described in more detail in this paper.

## 2. Survey of the Literature

Most of the research on fibre-cement boards to date has been devoted to the effect of operational factors [13,14,15] and the effect of high temperatures, investigated by testing the physicomechanical parameters of the boards, mainly their bending strength (*MOR*). The paper by Ardanuy et al. [6] presents the results of research on, inter alia, the effect of high temperature on fibre-cement boards, but only with reference to bending strength *MOR*. Li et al. [16] studied the effect of high temperatures on composites produced using the extrusion method, but also solely on the basis of their mechanical properties. The nondestructive investigations of fibre-cement boards have been mainly limited to the detection of imperfections arising at the production stage. Papers by Drelich et al. [8] and Schabowicz and Gorzelańczyk [17] presented the possibility of using Lamb waves in a noncontact ultrasonic scanner to detect defects in fibre-cement boards at the production stage. Paper [18] by Stark describes a method of detecting delaminations in composite elements by means of a moving ultrasonic probe. An ultrasonic device and a way of detecting delaminations are described in work [7] by Dębowski, Lewandowski, Mackiewicz, Ranachowski and Schabowicz. In works by Berkowski et al. [19], Hoła and Schabowicz [20] and Davis et al. [21] it was proposed to use the impact-echo method jointly with the impulse response method to identify delaminations in concrete members. Since the impulse response method is used to test elements thicker than 100 mm, it is not suitable for testing fibre-cement boards. Preliminary research had shown that striking such a board with a hammer could damage it, which is another reason why the impulse response method is not suitable for fibre-cement boards. Also, the impact echo method is not used to test fibre-cement boards. The drawback of this method is that multiple reflections of waves cause disturbances, making the interpretation of the obtained image difficult [19]. Therefore, it is not recommended to use this method for fibre-cement boards thicker than about 8 mm. 

In the literature, there is little information on the use of other nondestructive methods for testing fibre-cement boards. The research described in the works of Chady et al. [9] and Chady and Schabowicz [22] showed the terahertz (T-Ray) method to be suitable for testing fibre-cement boards. Terahertz signals have a very similar character to that of ultrasonic signals, but their interpretation is more complicated. In work [23], the microtomography method was used to identify delaminations and low-density regions in fibre-cement boards. The test results indicate that this method clearly reveals differences in the microstructure of the boards. Therefore, the microtomography method can be a useful tool for testing the structure of fibre-cement boards in which defects can arise due to production errors. However, this method can be used only for small-sized boards. It should be noted that, so far, few cases of testing fibre-cement boards by means of an acoustic emission have been reported in the literature. Ranachowski and Schabowicz et al. [23] carried out pilot studies of fibre-cement boards produced using the extrusion method, including boards subjected to the temperature of 230 °C for 2 h, in which the acoustic emission method was used to determine the effect of cellulose fibres on the strength of the boards and where attempts were made to distinguish between the AE events emitted by the fibres and the cement matrix. The results of this research confirmed the suitability of this method for testing fibre-cement boards. In paper [11] by Gorzelańczyk et al., it was proposed to use the acoustic emission method to investigate the impact of high temperature on fibre-cement boards. It should be noted that the effects of high temperatures on concrete, and the interdependences involved in this process, have been widely described using the acoustic emission method; examples are the works [24,25] by Ranachowski. One should also note that the acoustic emission method is often used to test thin materials, for example steel and polymeric composites, and even to test brittle food products [26,27]. During acoustic emission measurements a lot of data are recorded, which need to be properly analysed and interpreted. For this purpose, it can be highly effective to combine the acoustic emission method with artificial intelligence, including artificial neural networks (ANNs). In the literature, ANNs are used to analyse and recognise the signals registered during the failure of various materials [28]. In paper [29], Schabowicz used ANNs to analyse the results of nondestructive tests carried out on concrete. Łazarska et al. [30] and Woźniak et al. [31] successfully used the acoustic emission method and artificial neural networks to analyse its results. Rucka and Wilde [32,33] successfully used the ultrasonic method to investigate damage to concrete structures.

Considering the above information, it was decided that the acoustic emission method combined with artificial neural networks would be proper for identifying the degree of degradation of fibre-cement boards exposed to fire.

## 3. Strength Tests

The effect of fire was investigated for two series of fibre-cement boards, designated respectively A and B. The basic parameters of the two series are specified in Table 1.

The effect of fire on the fibre-cement boards was investigated by applying a flame generating a temperature of about 400 °C to the surface of the board, for about 1–15 min, every 2.5 min. The reference samples under an air-dry condition were designated as A and B. The designations of the other series of boards are listed in Table 2.

Since not the whole fire separation area, but the fibre-cement board alone was the object of the test, on the basis of experiments the duration of exposure to fire was limited to 15 min. A stand for exposing the samples to fire at the temperature of 400 °C is shown in Figure 2.

In order to identify the effect of the high temperature generated by fire on the fibre-cement boards, acoustic emission investigations were carried out during three-point bending. Before the investigations, the samples were exposed to fire, then they were placed in a strength testing machine and subjected to three-point bending. In the course of the three-point bending, the breaking force F, the strain ε and acoustic emission signals were registered. The tested samples are shown in Figure 3, while the stand for three-point bending tests and the set of equipment for acoustic emission measurements are shown in Figure 4.

The trace of flexural strength σ_m_, bending strength *MOR*, notch toughness W_f_, limit of proportionality (*LOP*) and strain ε were analysed. Bending strength *MOR* was calculated from the standard formula [34]:(1)$$MOR\;=\;\frac{3Fl_{s}\;}{2b\;e^{2}}$$
where:*F* is the loading force (N);*l_s_* is the length of the support span (mm);*B* is the specimen width (mm); and*e* is the specimen thickness (mm).

Notch toughness was calculated from the following formula given in [35]:(2)$$W_{f}=\frac{1}{S}\overset{F0,4max}{\underset{F0}{\int }}F\;da$$
where:*F* is the loading force (N);*S* is the specimen cross-section (m2); and*a* is the specimen deflection under the loading roller (m).

Figure 5 shows diagrams of the σ−ε dependence under bending for the samples of all the tested fibre-cement boards.

The results presented in Figure 5 show that, under high temperature, the bending strength *MOR* initially increases by 3–15% and then sharply decreases in comparison with that of the reference fibre-cement boards. In the initial phase of exposure to fire, especially in the time interval of 1–2.5 min, the sample would dry, but no fibres would be destroyed yet. It should be noted that the reference samples were under an air-dry condition and their bulk moisture content amounted to 6–8%. One can suppose that the dampness of the fibre-cement boards affects the bending strength *MOR* mainly as a result of the weakening of the bonds between the crystals in the structural lattice of the cement matrix. The weakening is due to the fact that, as the material’s dampness increases, the bonds partially dissolve, whereby the bending strength slightly decreases. The decrease in bending strength *MOR* caused by dampness is partially reversible; i.e., after it is dried, the material regains most of its lost strength and so ultimately its strength is close to the initial one. In the case of the B_1_–B_8_ series samples, no significant changes in the σ−ε dependence diagrams for the reference boards and the ones exposed to fire were observed, except for the changes in bending strength values. In the case of the A series boards, the σ−ε dependence changed markedly with the time of their exposure to fire. On this basis, for the boards of series A, one can determine the effect of high temperature, reflected not only in the initial increase in bending strength *MOR*, followed by its decrease, but also in changes in the plot of σ−ε dependence. For the series A boards, it was also observed that as the time of their exposure to the temperature of 400 °C is extended, the structure of the fibre-cement board becomes stiffer and more brittle. For this series, as the time of exposure to the temperature generated by fire is extended, the extent of the nonlinear increase in bending stress is reduced until the bending strength *MOR* comes to be level with the proportionality limit *LOP*. In the case of the reference boards, the *MOR* and *LOP* values were clearly separated.

Figure 6 shows the obtained values of notch toughness W_f_ of the tested fibre-cement boards.

An analysis of the results presented in Figure 6 showed that under the impact of high temperature, the notch toughness W_f_ of the boards of series B initially increases by 2–10% and then markedly decreases. For the boards of series A, the notch toughness W_f_ decreases with the time of exposure to fire. Hence, one can conclude that, as a result of the high temperature, the fibre-cement board matrix stiffens and becomes more brittle. The investigations carried out for the boards of series B showed that the identification of the degree of degradation on the basis of bending strength *MOR*, notch toughness W_f_, proportionality limit *LOP* and σ−ε dependence is inadequate. The results obtained for the fibre-cement boards of series A indicate that destructive changes took place in their structure. However, the fact that the bending strength *MOR* increased in the initial phase of the exposure to the high temperature is worth noting. The above results show that the identification of the degree of degradation solely on the basis of mechanical parameters, such as *MOR* and W_f_, is insufficient to determine the destructive effect of high temperature on fibre-cement boards in the initial phase of their exposure to fire.

## 4. Tests Using the Acoustic Emission Method and Artificial Neural Networks

The next step in the investigations of the degree of degradation caused by fire consists in analysing the AE signals registered during three-point bending, such as the events rate *N_ev_*, the events sum ∑*N_ev_*, the events energy *E_ev_* and the frequency distribution of the AE signal. For AE measurements, a broadband sensor with a frequency band of 5–500 kHz and an AE signal analyser (made by IPPT PAN, Warsaw, Poland), were used [35]. In the AE analyser the signal from the sensor is amplified and prefiltered to remove the acoustic background from the surroundings of the monitored element. Then, the signal is converted into a digital form. Further processing of digital records was carried out using audio file analysis software (made by IPPT PAN, Warsaw, Poland). Figure 7 shows exemplary values of events sum ∑*N_ev_* for the boards of series A1–A6 and B1–B8. No three-point bending test was carried out for cases A6-A8 because the fibre-cement boards had been completely destroyed by fire.

In order to more closely analyse the course of the bending test and how it was affected by the degrading factor, in the form of the high temperature of 400 °C lasting from 1 to 15 min, the dependence between events rate ∑*N_ev_* and bending stress *σ_m_* over time for selected cases is presented in Figure 8.

An analysis of the obtained results showed that AE signals were registered only after the stress corresponding to bending strength *MOR* had been exceeded. The analysis of the course of the three-point bending test, based on AE events sum ∑*N*_ev_, confirmed the changes taking place in the fibre-cement boards. Marked qualitative changes in the registered events in comparison with the reference boards were visible. At the high temperature of about 400 °C, the sample is initially dried, whereby the strength of the board and the acoustic activity of the AE events registered during bending slightly increase. An analysis of the AE descriptors indicates that the brief effect of the high temperature of 400 °C proved to be significant for the fibre-cement boards. The graphs for series and A and B show a noticeable decrease in the registered events and a change in the path of events sum ∑*N*_ev_. Besides the decrease in the value of events sum ∑*N*_ev_ due to the exposure to the high temperature of 400 °C for 7.5 min in the case of series A_5_ and for 12.5 min in the case of series B_7_, it was also noticed that all the registered events were within one 1.5 second time interval. Therefore, it can be concluded that the events originated from a single fracture of the cement matrix. This indicates that the exposure to the high temperature of 400 °C for about 7.5 min was more destructive for the fibre-cement boards of series A. Because of the destructive effect of the high temperature of about 400 °C, the registered AE event counts declined.

By analysing the acoustic activity traces in the time-frequency system during the bending of the boards of series A and B exposed to the temperature of 400 °C for over 7.5 min, model (reference) acoustic spectra for the breaking of the cement matrix devoid of fibres were selected. A model characteristic of the fibre breaking acoustic spectrum was selected from the spectra read for the boards of series A_1_ to B_1_ under an air-dry condition, with a similar characteristic repeating itself in the frequency range of 10–24 kHz, clearly standing out against the characteristic of the cement matrix. The characteristic of the background acoustic spectrum originating from the strength tester was determined on the basis of the initial bending phase by averaging the characteristics obtained from all of the tested fibre-cement boards of series A and B. The selected spectral characteristics of fibre-breaking are understood to be the signal accompanying the breaking of the cement matrix together with the fibres, whereas the model matrix spectral characteristic is understood to be the signal accompanying the breaking of the matrix alone. The selected model (reference) acoustic spectrum characteristics were recorded in 80 intervals at every 0.5 kHz. Figure 9 shows the record of the model characteristics of the acoustic spectrum accompanying the breaking of, respectively, the cement matrix and the fibres, and of the background acoustic spectrum.

According to Figure 9, the acoustic activity of the background is in the range of 10–15 dB. The characteristic of the cement matrix acoustic spectrum reached the acoustic activity of 25 dB in the range of 5–10 kHz (segment 1) and 20–32 kHz (segment 3). In the case of the fibres, the activity was above 25 dB in the frequency range of 12–18 dB (segment 2) and 32–38 kHz (segment 4). The above reference spectra for the cement matrix, the fibres and the background were implemented in artificial neural networks (ANNs) for training and testing. A unidirectional multilayered ANN structure with error backpropagation with momentum was adopted. Eight appropriate learning sequences were adopted in an iterative manner to achieve optimal compatibility of the learned ANN with the training pattern. After the ANN had been trained on the input data, its mapping correctness was verified using the training data and the testing data. For this purpose, two pairs of input data were fed, i.e., the data used for training the ANN, to check its ability to reproduce the reference spectra, and the one used for testing the ANN, to check its ability to identify the reference spectral characteristics originating from the fibres and the cement matrix during the bending test. For the eight performed training sequences, ANN compliance was obtained with the training standard at 0.995. Then, a record of the ANN output in the form of recognised acoustic spectra for, respectively, fibre breaking, matrix breaking and the background was obtained.

Figure 10 shows the results of the acoustic spectrum pattern recognition for the cement matrix and the fibres. The results are marked on the record of events rate *N*_ev_ and flexural stress σ_m_ versus time. The graphs are for the reference samples of series A_1_ and the samples of series A_3_–A_5_ exposed to fire. In order to better distinguish between the recognised acoustic spectra, the matrix patterns were marked green while the ones for the fibres were marked in violet.

An analysis of the results presented in Figure 10 showed that the registered acoustic events repeatedly predominate after the bending strength *MOR* is exceeded. The recognised events result from the breaking of the fibres and the cement matrix. An event originating from cement matrix breaking initiates subsequent events originating from fibre breaking. The matrix fractures when the bending strength *MOR* is reached. Subsequently, events originating from the breaking of the fibres build up until the board’s cross-section fractures. In Figure 10c, which shows the recognised events for the boards of series A_4_ exposed to the high temperature for 5 min, one can clearly see that the drop in the counts of acoustic events originating from the fibres subjected to the high temperature is smaller. Similarly to the case of the boards of series A_3_, the fracture of the cement matrix initiates the breaking of the fibres. The graph of flexural stress *σ*_m_ has no segment characterised by the linear increase of stress relative to strain. The limit of proportionality coincides with the bending strength *MOR*.

For the series A_5_ sample, exposed to the high temperature of 400 °C for 7.5 min, only events recognised as originating from the breaking of the matrix were registered within the time of 0.1 s. All of the fibres in this board were destroyed. The samples of series A_6_–A_8_ exposed to fire for over 7.5 min underwent complete destruction, whereby no further tests could be carried out. Table 3 lists the events recognised as accompanying fibre breaking and cement matrix breaking for the fibre-cement boards of series A_2_–A_5_.

Figure 11 shows diagrams for samples of series B1, B5 and B8.

In Figure 11, one can see a decline in the counts of events recognized for the fibres and a considerably shorter time segment in which events were registered. It should be noted that, in the case of the fibre-cement boards of series B, distinct changes in the registered AE descriptors occurred only when the boards were exposed to the high temperature for 7.5 min. Moreover, the proportionality limit of the boards of series B was equal to their bending strength *MOR*. In Figure 11c, which shows the results for series B_8_ exposed to the high temperature of 400 °C for 15 min, one can see that the recognised events originate from cement matrix breaking, but there are no recognised events originating from the fibres.

Table 4 lists the events recognised as accompanying fibre breaking and cement matrix breaking for boards of series B_1_–B_8_.

The above investigations have shown that it is insufficient to determine the degree of degradation solely on the basis of the value of bending strength *MOR*. As part of the investigations, the degree of degradation was determined not only on the basis of bending strength *MOR*, but also using the number of the registered AE events recognised (by artificial neural networks) as accompanying fibre breaking, and the dependence between notch toughness *W*_f_ and the events recognised as originating from fibre breaking. In this way, the degree of degradation caused by the high temperature could be unequivocally identified. As the number of events recognised as fibre breaking decreases, the notch toughness W_f_ also commensurately decreases. The sharp drop in notch toughness W_f_ is connected with the large number of fibres contained in fibre-cement boards and their rapid degradation under high temperature. It was experimentally found that the coefficient *U*_i_ determines the significance of the degradation of fibre-cement boards caused by high temperature. The coefficient *U*_i_ is expressed by the formula:(3)$$U_{i}\;=\;\frac{W_{fi}}{W_{f,ref}}$$
where: *U*_i_ is a value of the notch toughness coefficient common for samples exposed to high temperature and for the reference samples, *W*_fi_ is the notch toughness of a fibre-cement board exposed to high temperature, and *W*_f,ref_ is the notch toughness of the reference board.

For the notch toughness coefficient *U*_i_, its average value *U*_av_ and the lower and upper limit of the confidence interval were determined. The confidence level was assumed to amount to 95% of the average value expressed by the formulas:(4)$$U_{L}\;=\;U_{av}-0.75\times \bar{s}$$
(5)$$U_{H}\;=\;U_{av}+0.75\times \bar{s}$$
where: *U*_L_ is the lower limit of the confidence interval (the confidence amounting to 95% of the average value of the notch toughness coefficient *U*), *U*_H_ is the upper limit of the confidence interval (the confidence amounting to 95% of the average value of notch toughness coefficient *U*), *U*_av_ is the average value of the notch toughness coefficient, and $\bar{s}$ is the standard deviation.

*U*_H_ = 0.85 was adopted as the boundary value of the upper limit of the confidence interval.

Hence, insignificant degradation is characterised by an insignificant decrease in registered events recognised as accompanying fibre breaking or its absence, in comparison with the reference fibre-cement boards. An insignificant decrease in events recognised as accompanying fibre breaking is such a decrease for which the notch toughness W_f_ satisfies the condition: *U*_L_ > 0.85. At the same time, the bending strength *MOR* condition, defined as *R*_L_ > 0.75, must be satisfied. Significant degradation is characterised by a significant drop in registered events recognised as accompanying fibre breaking, in comparison with the reference board. A significant drop in registered events recognised as accompanying fibre breaking is such a drop for which the notch toughness *W*_f_ satisfies condition *U*_L_ < 0.85. Simultaneously, the bending strength *MOR* condition, defined as *R*_L_ > 0.75, must be satisfied. Critical degradation is characterised by the total absence of or a significant drop in events recognised as accompanying fibre breaking, in comparison with the reference board, and when the fibre-cement boards do not satisfy the bending strength *MOR* condition, defined as *R*_L_ < 0.75.

Table 5 and Table 6 show the degrees of degradation identified for the tested samples of the fibre-cement boards of, respectively, series A and B.

## 5. Conclusions

High temperatures by nature have a degrading effect on most building products. The latter’s resistance to high temperatures is measured by the length of time during which they preserve the properties required by the standards. The tests carried out on the fibre-cement boards of series A and B showed that they differed in the length of time during which they were resistant to fire. It should be emphasised that the tests have shown that it not sufficient to determine the resistance to high temperature solely on the basis of bending strength *MOR*. An analysis of the events registered during the bending test and their assignment to the signals accompanying the breaking of the fibres or the cement matrix broaden the range of investigations whereby one can correctly determine the high-temperature resistance of fibre-cement boards. Thanks to the use of the nondestructive methods and artificial neural networks, the degree of degradation of the fibre-cement boards exposed to the high temperature generated by fire was successfully determined. The following three degrees of degradation: insignificant degradation, significant degradation and critical degradation, were defined on the basis of the test results. Owing to the formulation of the dependence between the notch toughness *W*_f_ and the events originating from fibre breaking, registered during three-point bending, the degree of degradation of the fibre-cement boards exposed to high temperatures could be properly parameterised. The dependence between the notch toughness *W*_f_ and the counts (∑*N*_ev,f_) of recognised events accompanying fibre breaking during the bending of the fibre-cement boards was used for this purpose.

## Figures and Tables

**Figure 1 materials-12-00656-f001:**
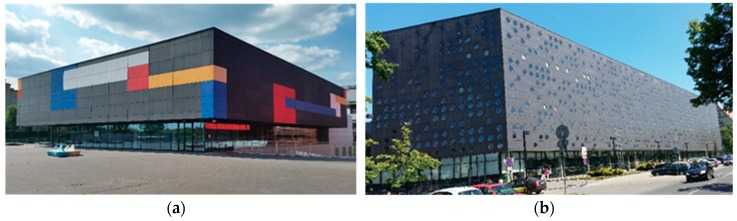
Exemplary uses of fibre-cement boards as ventilated façade cladding: (**a**) University building in Łódz, Poland, (**b**) University building in Wrocław, Poland.

**Figure 2 materials-12-00656-f002:**
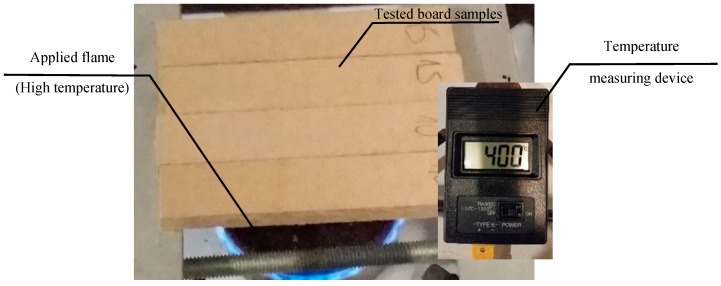
The stand for exposing samples to fire at the temperature of 400 °C.

**Figure 3 materials-12-00656-f003:**
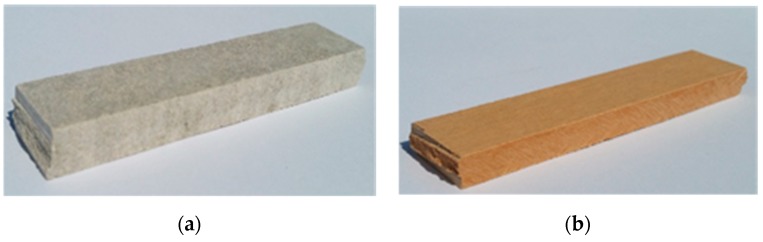
The tested fibre-cement board samples: (**a**) board A, (**b**) board B.

**Figure 4 materials-12-00656-f004:**
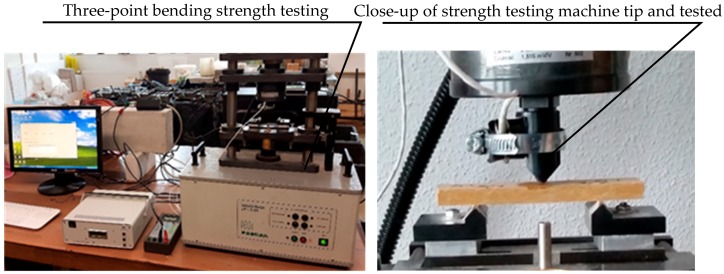
The test stand for the acoustic emission measurements, and a fibre-cement board during a test.

**Figure 5 materials-12-00656-f005:**
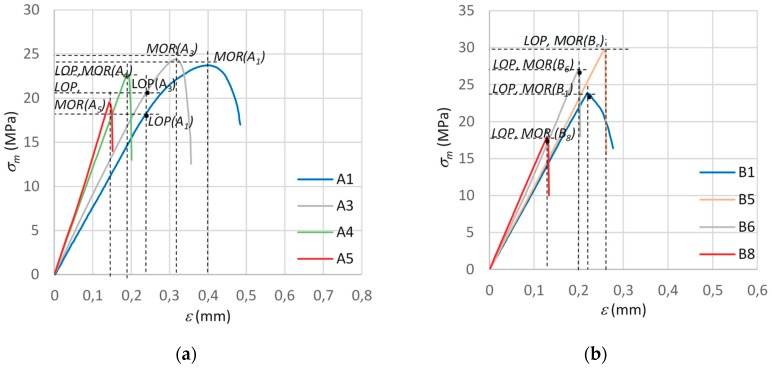
σ−ε dependence under bending for fibre-cement boards: (**a**) series A, (**b**) series B. *LOP*, limit of proportionality.

**Figure 6 materials-12-00656-f006:**
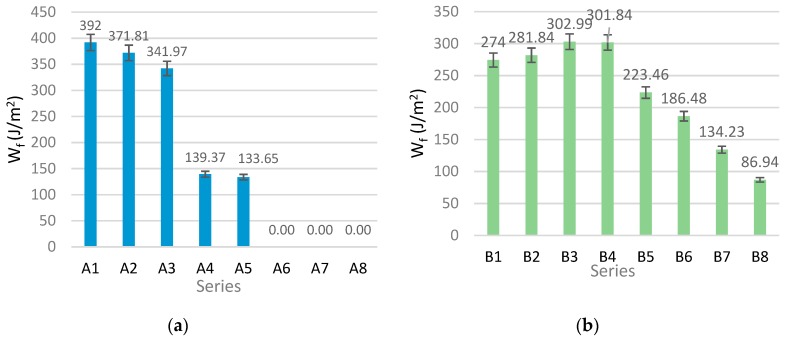
The values of notch toughness *W*_f_ for air-dry samples (case 1) and samples exposed to the temperature of 400 °C (cases 2–8): (**a**) series A, (**b**) series B.

**Figure 7 materials-12-00656-f007:**
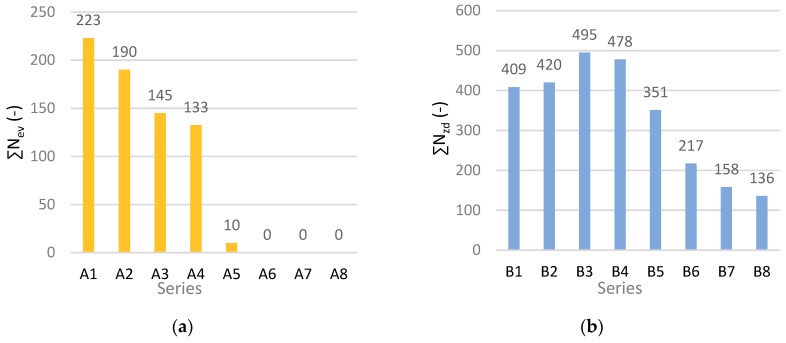
Exemplary registered values of events sum ∑*N*_ev_ for boards of series A under an air-dry condition (case 1) and exposed to the temperature of 400 °C (cases 2–8): (**a**) series A, (**b**) series B.

**Figure 8 materials-12-00656-f008:**
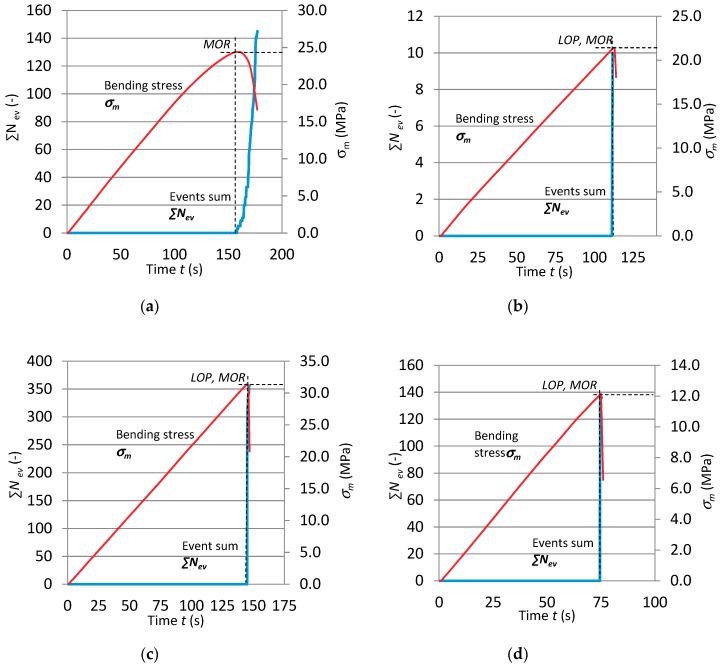
Dependence between σ_m_ and ∑*N*_ev_ as function of time *t* for fibre-cement boards: (**a**) series A_3_, (**b**) series A_5_, (**c**) series B_5_, (**d**) series B_8._

**Figure 9 materials-12-00656-f009:**
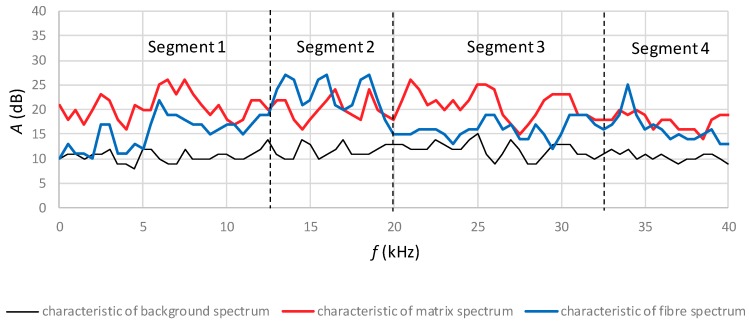
The characteristic of the background, cement matrix and fibre acoustic spectrum.

**Figure 10 materials-12-00656-f010:**
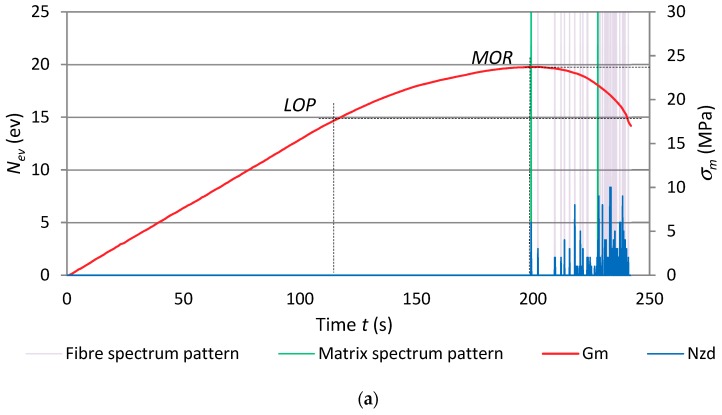
The events rate *N_ev_* and flexural stress σ_m_ versus time for exposure to fire, with marked recognised reference spectral characteristics: (**a**) series A_1_, (**b**) series A_3_, (**c**) series A_4_, (**d**) series A_5_.

**Figure 11 materials-12-00656-f011:**
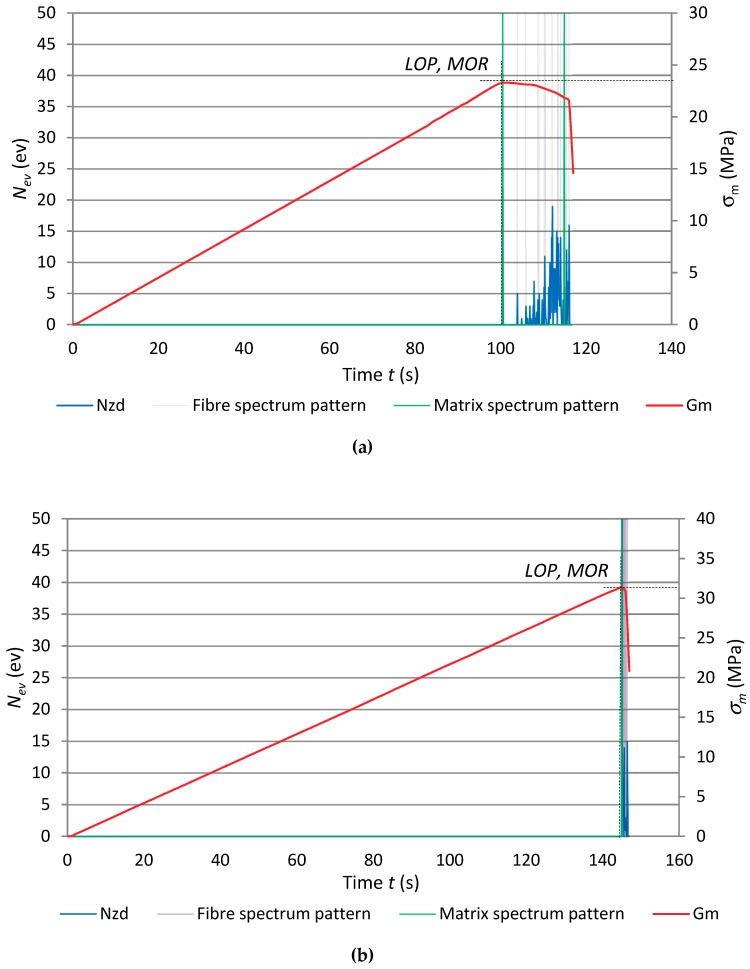
The events rate *N*_ev_ and flexural stress σ_m_ versus time for exposure to fire, with marked recognised reference spectral characteristics: (**a**) series B_1_, (**b**) series B_5_, (**c**) series B_8_.

**Table 1 materials-12-00656-t001:** The tested fibre-cement boards series A and B and their basic parameters.

Series Name	Board Thickness *e* (mm)	Board Colour	Application	Board Bulk Density *ρ* (g/cm^3^)	Bending Strength *MOR* (MPa)
A	8.0	natural	exterior	1.60	25
B	8.0	pigmented	exterior	1.60	30

**Table 2 materials-12-00656-t002:** A list of fibre-cement board series and test cases with the adopted sample designations.

Series Name/Test Case	SERIES A	SERIES B
Air-dry condition (reference board)	A_1_	B_1_
Exposure to the temperature of 400 °C: 1 min	A_2_	B_2_
Exposure to the temperature of 400 °C: 2.5 min	A_3_	B_3_
Exposure to the temperature of 400 °C: 5 min	A_4_	B_4_
Exposure to the temperature of 400 °C: 7.5 min	A_5_	B_5_
Exposure to the temperature of 400 °C: 10 min	A_6_	B_6_
Exposure to the temperature of 400 °C: 12.5 min	A_7_	B_7_
Exposure to the temperature of 400 °C: 15 min	A_8_	B_8_

**Table 3 materials-12-00656-t003:** A list of events recognised as accompanying fibre breaking and cement matrix breaking for boards of series A_1_–A_8_.

Series	Events Sum ∑*N*_ev_	Sum of Recognised Events ∑*N*_ev,r_	Sum of Events Ascribed to Fibre Breaking ∑*N*_ev,f_	Sum of Events Ascribed to Matrix Breaking ∑*N*_ev,m_
A_1_	223	203	195	8
A_2_	190	183	169	14
A_3_	145	144	70	70
A_4_	133	130	45	85
A_5_	92	92	0	92

**Table 4 materials-12-00656-t004:** A list of events recognised as accompanying fibre breaking and cement matrix breaking for boards of series B_1_–B_8_.

Series	Events Sum ∑*N*_ev_	Sum of Recognised Events ∑*N*_ev,r_	Sum of Events Ascribed to Fibre Breaking ∑*N*_ev,f_	Sum of Events Ascribed to Matrix Cracking ∑*N*_ev,m_
B_1_	409	405	325	80
B_2_	420	413	319	94
B_3_	495	484	336	148
B_4_	478	475	315	160
B_5_	351	349	198	151
B_6_	217	207	54	153
B_7_	158	158	0	158
B_8_	114	114	0	114

**Table 5 materials-12-00656-t005:** The identified degrees of degradation of the tested fibre-cement boards of series A.

Series	*R*_L_ (-)	*U*_L_ (-)	Degree of Degradation
A_2_	0.95	0.92	Insignificant degradation
A_3_	1.00	0.79	Significant degradation
A_4_	0.80	0.35	Significant degradation
A_5_	0.7	–	Critical degradation

**Table 6 materials-12-00656-t006:** The identified degrees of degradation of the tested fibre-cement boards of series B.

Series	*R*_L_ (-)	*U*_L_ (-)	Degree of Degradation
B_2_	0.96	0.99	Insignificant degradation
B_3_	0.94	1.00	Insignificant degradation
B_4_	1.00	1.00	Insignificant degradation
B_5_	1.00	0.81	Significant degradation
B_6_	0.91	0.68	Significant degradation
B_7_	0.56	–	Critical degradation
B_8_	0.54	–	Critical degradation

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
