# Peer review of "Identification of the Degree of Degradation of Fibre-Cement Boards Exposed to Fire by Means of the Acoustic Emission Method and Artificial Neural Networks"

_materials, 2019, doi:10.3390/ma12040656_

Reviewer 1 Report

The manuscript entitled "Identification of the degree of degradation of fibre-cement boards exposed to fire, by means of the acoustic emission method and artificial neural networks" contains original and significant results based on well conducted experiments and measurements. The results are well discussed and the conclusion is adequate to them. Therefore I recommend the manuscript for publication in Materials in the present form.

I am interested in the structure changes of fibre-cement boards related to the high temperature exposition. Therefore I recommend the authors to evaluate (e.g. SEM on fracture surfaces and cross sections) and describe the fibres nature and prepare next paper to correlate it with the results presented in this paper.

Author Response

REPLY TO REVIEWER’S COMMENTS                                                                      R#1

We are deeply grateful to the Reviewer for the effort put in the review of our paper.

We agree with most of the Reviewer’s comments and we have taken them into account in the paper’s revised version.

The authors are convinced that many of the Reviewer’s suggestions will be helpful in further research and analyses which will form the basis for the next paper.

Once again we would like to thank the Reviewer most warmly for the perceptive and detailed comments, which greatly enhance the understanding of the paper and its value.

Reviewer 2 Report

1. The paper lacks detailed explanation of the procedures and analyses conducted in the study. The clarity of the paper needs to be improved as well.

 For example:

a)      Probably include a brief discussion about the acoustic emission testing setup, what it measures, how it measures, the signal processing applied, etc.

b)      I think you should give a brief explanation to how the neural network is applied in acoustic emission testing? Discuss the input data, the output data, the data sets used for training and testing, and the verification process.

c)      Stating “exposed to fire for 1 to 15 mins” gives confusion to the readers. I think it is better to say that “the fibre-cement board samples were exposed to fire at various durations in the range of 1 to 15 mins”.

I think you should explain Figure 9 a little bit better since it is important in the ANN application?

2. Is it possible to determine the deterioration of the materials using acoustic emission (high temperature environment) while the board was applied with flame?

In the abstract, it focused too much on the gradual degradation of the fibers. This somehow gives readers the impression that the fibers were mainly the reason of the degradation of the cement boards. It was discussed in the paper that moisture content also contributed in the degradation of the fibre-cement boards. Furthermore, the breaking of the fibers occur as the cement breaks. Does the instantaneous breaking of the cement boards applied with flame at long durations, which show no event of fibers breaking, validate that the fibers were the main reason of the strength of the boards? The results in Figure 5 already show a decrease in strength even before the cement matrix breaks. Take note, that you also mentioned that at a temperature of 500 degrees Celsius the board was destroyed after 1-2 minutes. Thus, longer durations of exposure at a temperature of 400 degrees Celsius may have similar effects to that of boards exposed at a temperature of 500 degrees Celsius.

3. Some figures and its labels need to be improved.
For example: In Figure 2, change "exposure to fire" to "applied flame or heat"

Is the apparatus in Figure 4 really called a stand?
Maybe you can use arrows for Figure 8 to label the lines and points.
Is it even necessary to use "respectively" in the label of Fig. 9?

4. Some abbreviations in the paper were used without defining it first, such as using AE in the abstract without indicating that it means acoustic emission.

5. Some symbols were not defined in the paper, especially in Equations 1 and 2. Don’t you think it is better to put the symbols which are beside its definition in a parentheses?  For example in page 1, line 22: bending strength (MOR).

6. What is the basis of the graph in Figure 7 and why do the values in Figure 8 differ from Figure 7.

7. The paper needs serious English editing.

 For example:

a)      The paper uses a lot of has been, have been, had been etc.

On page 1, line20 – “It has been shown . . .” could be changed to “It was shown that it is not . . .”

On page 2, lines 66 – “Preliminary investigations showed that at a temperature of 250 degrees Celsius the fibres in the boards have been completely destroyed after 2-3 hours.”

b)      Page 1, Lines 14 to 15, abstract – Improve the sentence into something like this “The fibre-cement board samples were initially exposed to fire at various durations in the range of 1 to 15 mins. After which, the samples were subjected to three-point bending and were investigated using the acoustic emission method.”

c)      Page 1, Lines 28 to 29 – “The fibre-cement board is a building material that has been used in construction . . .”

d)      The sentence in page 1, lines 34-36 confuse me. Shouldn’t the sentence be broken into two starting at “whereby . . .”

e)      Several sentences lack comma such as in page 2, line 38 to 39. “In the course of their service life, fibre-cement boards . . .”; page 2, line 45 – “After building fire occurs, it is necessary . . . “. Please review the manuscript. Take note that I also added “occur” in the sentence.

f)       Improve the sentence in page 2, lines 58 to 60. Suggestion: “In order to verify this hypothesis, experiments were conducted during which fibre-cement board samples were exposed to fire at a temperature of about 400°C at various durations in the range of 1 to 15 mins.”

g)      Page 2, line 61 to 63 – “The aim of the experimental research is to observe the advancement of degradation of the fibres . . .”. However, the sentence does not somehow go in line with the methodology of the study since the degradation was only characterized after three-point bending and not during when the boards were exposed to fire. Maybe change it to “degradation of the fibres due to fire and associated high temperature.”

There are still so many sections of the paper which require English editing. Hoping that you improve the manuscript.

8. There are so many referencing errors in the paper. The way you use “et al.” and how you cite references in the text feels like an error to me. For example, you used “et al.” for a paper which has one or two authors.

Author Response

REPLY TO REVIEWER’S COMMENTS                                                                     

We are deeply grateful to the Reviewer for the effort put in the review of our paper.

We agree with most of the Reviewer’s comments and we have taken them into account in the paper’s revised version.

 1. The paper lacks detailed explanation of the procedures and analyses conducted in the study. The clarity of the paper needs to be improved as well.

 For example:

a)      Probably include a brief discussion about the acoustic emission testing setup, what it measures, how it measures, the signal processing applied, etc.

b)      I think you should give a brief explanation to how the neural network is applied in acoustic emission testing? Discuss the input data, the output data, the data sets used for training and testing, and the verification process.

c)      Stating “exposed to fire for 1 to 15 mins” gives confusion to the readers. I think it is better to say that “the fibre-cement board samples were exposed to fire at various durations in the range of 1 to 15 mins”.

I think you should explain Figure 9 a little bit better since it is important in the ANN application?

 This has been complied with

 2. Is it possible to determine the deterioration of the materials using acoustic emission (high temperature environment) while the board was applied with flame?

 In the abstract, it focused too much on the gradual degradation of the fibers. This somehow gives readers the impression that the fibers were mainly the reason of the degradation of the cement boards. It was discussed in the paper that moisture content also contributed in the degradation of the fibre-cement boards. Furthermore, the breaking of the fibers occur as the cement breaks. Does the instantaneous breaking of the cement boards applied with flame at long durations, which show no event of fibers breaking, validate that the fibers were the main reason of the strength of the boards? The results in Figure 5 already show a decrease in strength even before the cement matrix breaks. Take note, that you also mentioned that at a temperature of 500 degrees Celsius the board was destroyed after 1-2 minutes. Thus, longer durations of exposure at a temperature of 400 degrees Celsius may have similar effects to that of boards exposed at a temperature of 500 degrees Celsius.

Using the acoustic emission and ANNs, it is possible to determine the degree of the destruction of fiber-cement boards under the influence of high temperature. The interaction of the flame is of course a specific effect of high temperature. The authors carried out various types of high temperature impact in the range (230-500C), including high temperature plates eg. in a laboratory furnace. Convergent results were obtained as with the flame effect confirming the destructive influence of high temperature on the fibers contained in the plate. Long-lasting exposure to the flame (high temperature) results in complete degradation of the fibers. Boards containing undamaged fibers (reference board) obtained higher values of bending strength MOR. The influence of moisture is important and above all for the cement matrix. Wetted boards achieve lower values of bending strength MOR. The short-term effect of the temperature causes the plate to dry (without damage), hence the slight increase in the MOR compared to the reference plate.

3. Some figures and its labels need to be improved.
For example: In Figure 2, change "exposure to fire" to "applied flame or heat"

Is the apparatus in Figure 4 really called a stand?
Maybe you can use arrows for Figure 8 to label the lines and points.
Is it even necessary to use "respectively" in the label of Fig. 9?

 This has been complied with

 4. Some abbreviations in the paper were used without defining it first, such as using AE in the abstract without indicating that it means acoustic emission.

This has been complied with

 5. Some symbols were not defined in the paper, especially in Equations 1 and 2. Don’t you think it is better to put the symbols which are beside its definition in a parentheses?  For example in page 1, line 22: bending strength (MOR).

 This has been complied with

 6. What is the basis of the graph in Figure 7 and why do the values in Figure 8 differ from Figure 7.

 This has been complied with

 7. The paper needs serious English editing.

 The linguistic errors have been corrected. The paper has been checked by a sworn translator of the English language.

The authors are convinced that many of the Reviewer’s suggestions will be helpful in further research and analyses which will form the basis for the next paper.

Once again we would like to thank the Reviewer most warmly for the perceptive and detailed comments, which greatly enhance the understanding of the paper and its value.

Reviewer 3 Report

The article is well organized and conducted. The applied method is also interesting and does support the drawn conclusions, properly. I could not find major drawbacks in this work for the publication of Journal of Materials.

Author Response

REPLY TO REVIEWER’S COMMENTS                                                                   

We are deeply grateful to the Reviewer for the effort put in the review of our paper.

We agree with most of the Reviewer’s comments and we have taken them into account in the paper’s revised version.

The authors are convinced that many of the Reviewer’s suggestions will be helpful in further research and analyses which will form the basis for the next paper.

Once again we would like to thank the Reviewer most warmly for the perceptive and detailed comments, which greatly enhance the understanding of the paper and its value.

Round  2

Reviewer 2 Report

Just some minor changes that are needed to be done.

Page 3, line 87 - [13, 14, 15]

Page 3, line 93-94 - "Papers by Drelich et al. [8] and Schabowicz and Gorzelańczyk [17] presented the possibility of . . ."

Page 3, line 108-109 - "The research described in the works of Chady et al. [9] and Chady and Schabowicz [22] showed that the . . ."

Are you using "latter" correctly?

Page 9, line 246 - bending stress "m"? "sigma_m"?

Author Response

We are deeply grateful to the Reviewer for the effort put in the review of our paper.

We agree with most of the Reviewer’s comments and we have taken them into account in the paper’s revised version.

Once again we would like to thank the Reviewer most warmly for the perceptive and detailed comments, which greatly enhance the understanding of the paper and its value.

The linguistic errors have been corrected. The paper has been checked by a sworn translator of the English language